# Sprint and Jump Mechanical Profiles in Academy Rugby League Players: Positional Differences and the Associations between Profiles and Sprint Performance

**DOI:** 10.3390/sports9070093

**Published:** 2021-06-25

**Authors:** Ben Nicholson, Alex Dinsdale, Ben Jones, Kevin Till

**Affiliations:** 1Carnegie Applied Rugby Research (CARR) Centre, Carnegie School of Sport, Leeds Beckett University, Leeds LS6 3QS, UK; A.Dinsdale@leedsbeckett.ac.uk (A.D.); B.Jones@leedsbeckett.ac.uk (B.J.); k.till@leedsbeckett.ac.uk (K.T.); 2Leeds Rhinos Rugby League Club, Leeds LS5 3BW, UK; 3School of Science and Technology, University of New England, Armidale, NSW 2351, Australia; 4Division of Exercise Science and Sports Medicine, Department of Human Biology, Faculty of Health Sciences, The University of Cape Town, Cape Town 7700, South Africa

**Keywords:** strength and conditioning, speed, acceleration, running, youth

## Abstract

This cross-sectional study evaluated the sprint and jump mechanical profiles of male academy rugby league players, the differences between positions, and the associations between mechanical profiles and sprint performance. Twenty academy rugby league players performed 40-m sprints and squat jumps at increasing loads (0–80 kg) to determine individual mechanical (force-velocity-power) and performance variables. The mechanical variables (absolute and relative theoretical maximal force-velocity-power, force-velocity linear relationship, and mechanical efficiency) were determined from the mechanical profiles. Forwards had significantly (*p* < 0.05) greater vertical and horizontal force, momentum but jumped lower (unloaded) and were slower than backs. No athlete presented an optimal jump profile. No associations were found between jump and sprint mechanical variables. Absolute theoretical maximal vertical force significantly (*p* < 0.05) correlated (r = 0.71–0.77) with sprint momentum. Moderate (r = −0.47) to near-perfect (r = 1.00) significant associations (*p* < 0.05) were found between sprint mechanical and performance variables. The largest associations shifted from maximum relative horizontal force-power generation and application to maximum velocity capabilities and force application at high velocities as distance increased. The jump and sprint mechanical profiles appear to provide distinctive and highly variable information about academy rugby league players’ sprint and jump capacities. Associations between mechanical variables and sprint performance suggest horizontal and vertical profiles differ and should be trained accordingly.

## 1. Introduction

The capabilities of a rugby league athlete’s neuromuscular system to produce and apply high mechanical power (force x time) are of great importance during the high-intensity movement demands of the sport (e.g., sprinting, jumping, and tackling) regardless of playing position [1,2,3,4,5,6,7,8]. Mechanical profiling is a method of assessing the maximum capabilities of an individual’s neuromusculoskeletal system (i.e., force-velocity-power and force-velocity relationship), which underpins athletic performance [2,9,10,11]. These are commonly calculated in jumping and sprinting tasks [9,10,11]. Mechanical profiling expands beyond traditional strength or power testing, incorporating the entire force-velocity spectrum [2]. For example, sprint mechanical profiling identifies distinctive capabilities, such as the maximum sprint velocity (v_max_), acceleration (a_max_), and acceleration relative to a time constant (τ), achieved to derive the mechanical variables beyond sprint split times alone. Thus, mechanical profiling provides valuable diagnostic information for athlete profiling, monitoring, and programming [2,7]. Within rugby league, a plethora of studies have evaluated the physical qualities of rugby league players (e.g., [12,13]), however, only three studies have presented the mechanical profiles from jumping [14] and sprinting [15,16] in senior elite rugby league cohorts. To date, no study in rugby league has included both methods in one cohort, profiled academy rugby league players, or provided comparisons between playing positions. Such information would provide added value for rugby league practitioners beyond maximum strength, jump height, and sprint times alone [12,13].

When allocating time and resources to assessing athletes’ mechanical capabilities, it is important to consider which testing methods can provide meaningful data to influence practice and transfer to on-field performance (i.e., sprint performance). In rugby league, sprinting performance (e.g., time for a given distance) has been shown as a differentiating factor between performance standards [17,18,19,20] and demonstrates moderate associations (r = 0.31 to 0.44) with attacking and defensive performance indicators. This is particularly important when sprint performance is assessed relative to body mass, represented as sprint momentum (body mass × velocity) given the game’s high collision requirements [21]. With both jump and sprint mechanical profiles exploring the lower limb’s maximal capabilities, it is important to (1) understand if they provide distinctive information between methods and (2) evaluate which variables have the largest associations with performance. Previous research has reported inconsistent associations between matched jump and sprint profiling variables across a range of populations (e.g., performance standards, sports, genders, [22,23,24]). Associations between matched mechanical profiling variables would suggest that (1) individual requirements could be inferred from either testing method; and (2) increases in jump variables would enhance sprint variables (i.e., increases in squat strength transferring to horizontal force and vice versa). To date, it is unclear the interchangeability of jump and sprint mechanical profiling methods and their associations with performance variables in youth athletes in development programmes (i.e., academy systems). Previous studies have only included senior populations or categorised athletes by playing standard with no identification of training experience, age of participants, or playing position [22,23]. This has important implications for the training practices in academy rugby players (i.e., training prescription on monitoring methodology).

While an optimal and force-velocity relationship exists in jumping and can be used as an individual reference for individualised training, there is currently no optimal value for sprint mechanical profiles for performance [2,25]. Instead, the associations of the mechanical variables have been shown to change depending on the distance outcome [22,23,26]. Currently, the literature has evaluated the association between mechanical variables with sprinting performance across a limited range of cumulative distance outcomes (e.g., 0–20 and 0–40 m times [22,23,26]). By assessing the associations between sprint mechanical profiles across a larger range of sport-specific sprint performance outcomes (e.g., split times and momentum measures), researchers and practitioners would have improved insight into the utility of the mechanical profiles in academy rugby league players. It would also provide a structure for a proposed hierarchy of training attention for targeted training prescription, athlete profiling, and monitoring to the individual needs of each athlete.

Therefore, the first aim of this study was to present and compare the jump and sprint mechanical profiles of male academy rugby league players between positions (i.e., forwards and backs). Second, the study aimed to investigate the relationships between (1) jump and sprint mechanical variables and (2) jump and sprint mechanical variables with sprint performance. Based upon the available rugby literature [7,12,23] it was hypothesized that (1) there would be significant positional differences in mechanical profiles between forwards and backs; (2) there would be no significant associations between jump and sprint mechanical variables and (3); sprint but not jump mechanical variables would be significantly associated with sprint performance.

## 2. Materials and Methods

### 2.1. Design

A cross-sectional research design was used. Jump and sprint mechanical profiles were assessed in academy rugby league players using the valid and reliable methods proposed by Samazino and colleagues [27,28,29,30,31]. These methods included maximal jumps trials with external load (0, 20, 40, 60, and 80 kg), 40 m sprint tests, anthropometric and environmental measurements. All data collection was completed in the last quarter of the competitive season on a single day (4.00–5.30 p.m.) with players encouraged to rest 48 h beforehand to reduce possible interference caused by fatigue.

### 2.2. Participants

Twenty male super league academy rugby league players (age 17.6 ± 0.9 years; height 179.9 ± 6.6 cm; body mass 91.2 ± 11.8 kg) from the same club participated in the study. All players trained on a part-time basis, participating on average in ~9 h of combined in-season rugby-specific training and competition per week (2–4 rugby training sessions, 1–3 resistance training sessions, 1–2 sprinting sessions, 1 domestic game per week). Sprinting sessions consisted of combinations of sprint technical drills (i.e., 10–30 m per drill) and various sprints distances (e.g., 10–50 m) and start positions, including curved sprints and partner band resisted sprints (total sprint volume ~150–300 m). Lower body resistance training consisted of a concurrent programme of power (e.g., 3 sets of 4–6 reps; unloaded/loaded jumps and weightlifting derivatives), strength (e.g., 3–6 sets of 3–6 repetition maximum (RM)), and hypertrophy (e.g., 3–6 sets of 8–15 RM) exercises (e.g., squat, deadlift, hip thrust, variations).” All players had >2 years prior structured strength and conditioning training experience and were highly familiar with the testing procedures as both sprints and loaded squat jumps were consistent components of their training programmes. Participants were free from any existing musculoskeletal injuries that would be aggravated by the exercises/testing. Participants were screened by medical staff to confirm that they were free of any lower-extremity musculoskeletal or neuromuscular injuries that would have affected their ability to perform the required loaded squat jump and sprinting task at a maximal effort. All participants received a clear explanation of the study and provided written consent to participate. Ethics approval was granted by the Leeds Beckett University ethics committee (Ref: 51872; 21 October 2018).

### 2.3. Procedures

All participants were instructed to rest the day before testing, to attend testing in a fed and hydrated state, with adequate sleep, similar to their normal practices before training. Participants completed a general warm-up consisting of jogging, low intensity running technique drills (i.e., 10 m × 2 marching and A-skips), lower-limb dynamic stretching (i.e., 6 reps each of lunge with torso rotation, bodyweight squat and hinge patterns, 10 m of high knees) and plyometrics (10 m skipping and bounding). The specific warm-ups comprised of both unloaded (×5) and loaded (×3 at +40 kg) squat jumps and progressively faster 40 m sprints (70%, 80%, and 90% of the subject’s self-perceived maximal velocity). Thirty seconds were provided between warm-up jump trials and 2 min rest between sprint and jump sets. All participants completed 3 min passive recovery before testing commenced, during which the testing procedures were verbally re-explained. The warm-ups were supervised by the club’s strength and conditioning staff.

#### 2.3.1. Squat Jump Testing Procedures

Participants performed two maximal jump trials with an external load of 0, 20, 40, 60, and 80 kg. As the load-velocity relationship in a squat jump has been shown to be linear, relative loads were not required [3,27]. The loaded jumps were performed with free-weight barbells (20 kg) in the high bar position or with arms crossed on the torso for the unloaded condition. Before each jump, participants were instructed to stand up straight. From this position, participants descended into a self-selected paused squat jump start position (~90–100° knee angle; 2 s hold), followed by a jump for maximum height [30]. Squat jump height was obtained using an OptoJump optical measurement system (Microgate, Bolzano, Italy). During each jump, the investigator signalled when to jump after the pause in the jump start position through verbal cueing (“hold, 2, 3, jump”). All jumps were completed on a wooden weightlifting platform. Two valid trials were performed with each load with 2–3 min recovery between trials and 4–5 min between loads condition. If an incorrect take-off or landing technique was observed (i.e., countermovement, extended leg in foot plantar flexion), the trial was excluded from calculations, and the trial was repeated. Both trials were used for within-day test re-test reliability measurements (intraclass correlation coefficient [ICC], typical error, and coefficient of variation [CV]). The within-day test re-test reliability data for squat jump heights across loaded and unloaded conditions was ICC = 0.68–0.90, typical error = 0.7–1.9 cm and CV = 2.4–5.7%. Only body weight (ICC = 0.73) and SJ 40 kg (ICC = 0.68) failed to demonstrate an acceptable (ICC > 0.8) [32]. Given the low %CV, the low ICC value can likely be explained by a low between-subject variance. Thus, despite the low ICC values, it cannot be excluded that the test-retest reliability is also high in this population [33].

#### 2.3.2. Sprint Testing Procedures

Participants performed two maximal 40 m sprints trials with 4–5 min rest between repetitions on an artificial-turf surface (3G). The distances were chosen to assess initial and maximal sprint capabilities as used by previous research [15,26,34]. The sprints were recorded using a radar gun device (Stalker ATS II, Applied Concepts, Dallas, TX, USA), which obtained forward sprinting instantaneous velocity-time-position data at 46.9 Hz. Participants initiated from a standing split-stance position with their preferred lead foot forward behind the start line. Once participants were in the start position, radar data capture was started, and participants could begin sprinting at any time (i.e., running times do not include a reaction time). The radar gun was mounted to a tripod positioned 3 m behind the starting line at the height of 1 m above the ground (corresponding approximately to the participant’s centre of mass. No false step was allowed at the start, and participants were instructed to provide maximal effort throughout each sprint trial. All participants wore studded training shoes (i.e., football boots) and team training attire. Both trials were used for within-day test re-test reliability measurements ICC, typical error, and CV. The within-day test re-test reliability data for sprint displacement-times was ICC = 0.75–0.96, typical error = 0.03–0.06 s and CV = 0.7–1.7% with reliability measures improving as sprint distance increased. Only 0–2 m time (ICC = 0.77) failed to demonstrate an acceptable test-retest reliability (ICC > 0.8) [32]. As with the jump data, the low %CV the low ICC value can likely be explained by a low between-subject variance. Thus, it cannot be excluded that the test-retest reliability is also high in this population [33].

#### 2.3.3. Anthropometric, Position and Environmental Measures for Mechanical Profile Calculations

The anthropometric (squat jump starting height (m), extended lower limb length (m), body mass (kg) and height (cm)) and environmental (barometric pressure (mmHg), air temperature (°C), and wind velocity (mph)) measurements were taken as previously identified [27,28] for the calculation of the force and velocity data.

### 2.4. Data Analysis

The raw data files were manually processed in the STATS software (STATS; Stalker ATS II Version 5.0.2.1; Applied Concepts, Dallas, TX, USA) as described by Simperingham et al., [35].

#### 2.4.1. Mechanical Profile Computation

The force and velocity data derived from the peak jump height for each of the five loads and the radar guns processed instantaneous velocity-time-position data from the fastest 40 m trial. The force and velocity data were then used to obtain for each subject (i) the individual slope of the inverse linear force-velocity relationship (F-v) and (ii) the individual second-degree polynomial power-velocity (P-v) relationship to calculate theoretical maximal horizontal force, velocity, and power capabilities respectively [25,27]. The mechanical effectiveness of force application during sprinting was quantified over each support phase or step by the ratio of horizontal force to the corresponding resultant ground reaction force (RF, in %), and over the entire acceleration phase by the slope of the linear decrease in RF when velocity increases (D_RF_, in % [2,4]. The optimal slope of the F-v for jump performance and the F-v imbalance (magnitude of the difference of the modelled profile from optimal) described by [3]. Qualitative interpretations of the F-v imbalance were provided for the jump mechanical profiles: high force deficit ≤ 60%, low force deficit = 60–90%, well balance ≥ 90–110%, low velocity deficit = 110–140% and high velocity deficit ≥ 140% based upon [1]. Currently, there is no optimal sprint mechanical for performance available. Therefore, comparisons in imbalances were limited to jump profiles only [25]. The calculations were completed in a custom-made Microsoft Excel spreadsheet [36,37]. For a complete overview of the formulas and data processing approach, see [27] for jump data and [28,29] for the sprint data.

#### Squat Jump Mechanical Variables

The jump mechanical variables recorded were; theoretical maximal force (F_V0_), theoretical relative (per kg of body mass), maximal force (F_V0rel_), theoretical maximal extension velocity (v_V0_), maximal mechanical power (P_Vmax_), maximal relative (per kg of body mass) mechanical power (P_Vmaxrel_), the slope of the linear F-v relationship (S_VFv_), the magnitude of the difference between actual and optimal F-v relationship known as the F-v imbalance (Fv_imb_) and squat jump heights for each load.

#### Sprint Mechanical Variables

The sprint mechanical variables recorded were theoretical maximal force (F_H0_), theoretical relative maximal force (F_H0rel_), theoretical maximal extension velocity (v_H0_), maximal mechanical power (P_Hmax_), maximal relative mechanical power (P_Hmaxrel_), the slope of the linear F-v relationship (S_HFv_). The mechanical effectiveness variables of force application were quantified using the maximal ratio of force (RF_max_) and the rate of decrease in the ratio of force (D_RF_).

#### 2.4.2. Sprint Performance Data

The sprint performance descriptors were derived from the custom-made Microsoft Excel spreadsheets [36,37] and the processed instantaneous velocity-time measurements. These included: sprint split times (0–2, 0–5, 0–10, 0–20, 0–30, and 0–40 m (s)) between split times (∆10–20, ∆20–30, and ∆30–40 m (s)), sprint momentum (0–10 and 30–40 m momentum (kg·s^−1^) and velocity trace characteristics (i.e., maximum sprinting velocity (v_max_), maximum sprint acceleration (a_max_) and acceleration relative to a time constant (τ)). The specified distances were chosen to enable the assessment of initial and maximal sprint capabilities as used by previous research [15,26,34].

### 2.5. Statistical Analysis

Descriptive statistics (group and positional mean and standard deviation, range, or frequency where specified) for the jump and sprint mechanical profiles were calculated. All variables were assessed for normality via the Shapiro-Wilk test. Statistical significance was set at an alpha level size of *p* ≤ 0.05 for all measured variables. Normally distributed mechanical and sprint performance results were compared between positions using independent samples *t*-tests and Cohen’s d effect size (ES) with 90% confidence intervals (CI). In instances of non-normally distributed data, the Mann-Whitney test was used. Effects sizes were interpreted as trivial (<0.2), small (0.20–0.59), moderate (0.60–1.19), and large (1.20–1.99), very large (2.0–3.99) and extremely large (>4.0) [32]. To analyse associations between jump and sprint mechanical profiles and between mechanical variables with sprint performance, all subject data (n = 20) was used. Normally distributed mechanical and sprint performance results were tested through Pearson’s correlation coefficients. In instances of non-normally distributed data, the Spearman rank test was used. Qualitative interpretations of the r coefficients were provided: trivial (r < 0.1), small (r = 0.1–0.3), moderate (r = 0.3–0.5), large (r = 0.5–0.7), very large (r = 0.7–0.9), and nearly perfect (r > 0.9) [32]. All analyses were performed using IBM Statistical Package for the Social Sciences (Version 20.0, SPSS for Windows Chicago, IL, USA).

## 3. Results

### 3.1. Descriptive Statistics

Table 1 presents the jump and sprints mechanical profiles and sprint performance outcomes of the academy rugby league players according to playing position.

### 3.2. Positional Differences

Forwards had significantly greater absolute vertical (large ES = 1.34 (2.17 to 0.38)) and horizontal (moderate ES = 1.06 (1.83 to 0.20)) force and had momentum (Large ES, 0–10 m = 1.73 (2.53 to 0.77) and 30–40 m = 1.56 (2.35 to 0.63)) than backs. Backs were significantly faster (moderate ES) than forwards over the ∆10–20 m (ES = −1.05 (−1.82 to −0.19)), and ∆30–40 m split times (ES = −0.95 (−1.72 to −0.11), jumped higher (unloaded, large ES = 0.96 (0.05 to 1.77)). No other significant differences were identified between positions; however, moderate differences were found for relative vertical power (ES = 0.63 (−0.24 to 1.44), absolute horizontal power (ES = −1.06 (−1.83 to −0.20), actual (ES = 1.11 (0.25 to 1.88) and theoretical maximal sprinting velocity (ES = 1.16 (0.28 to 1.93) and split times ≥ 20 m (0–20 m (ES = −0.66 (−1.43 to 0.15), 0–30 m (ES = −0.69 (−1.46 to 0.13), 0–40 m (ES = −0.88 (−1.65 to −0.04), ∆20–30 m (ES = −0.94 (−1.71 to −0.10)). The jump mechanical profiles represented a range of high-low force deficit profiles in backs and high-low force and velocity deficits and balanced profiles in forwards (Table 1). A force deficit was most common in the participants for jump profiles (15/18).

### 3.3. Associations between Jump and Sprint Mechanical Variables

Figure 1 shows the associations between the body mass relative to matched jump and sprint mechanical variables. Trivial-small positive, non-significant relationships (r = 0.09–0.28) were found between the body mass relative matched vertical and horizontal mechanical variables.

#### 3.3.1. The Association between Relative Theoretical Maximal Force Variables

The trivial association between vertical and horizontal relative theoretical maximal force variables failed to reach statistical significance (r = 0.09, *p* = 0.73).

#### 3.3.2. The Association between Velocity Variables

The small association between vertical and horizontal theoretical maximal velocity variables failed to reach statistical significance (r = 0.28, *p* = 0.26).

#### 3.3.3. The Association between Relative Theoretical Maximal Power Variables

The small association between vertical and horizontal relative theoretical maximal power variables failed to reach statistical significance (r = 0.27, *p* = 0.28).

#### 3.3.4. The Association between the Slope of the Linear F-v Relationships

The small association between the vertical and horizontal slope of the linear F–v relationships failed to reach statistical significance (r = 0.24, *p* = 0.34).

### 3.4. Associations between Jump and Sprint Mechanical Profiles and Sprint Performance Outcomes

Table 2 shows the association between jump mechanical profiles and sprint performance outcomes. Very large (r = 0.71–0.75) significant positive associations (*p* < 0.01) were found between F_V0_ and both momentum outcomes. All other mechanical variables were non-significant with no to moderate associations found (r = 0.00–0.41).

Table 3 shows the association between sprint mechanical variables and each of the sprint performance outcomes. Moderate to near-perfect significant relationships (positive and negative) were found between sprint mechanical profiles and sprint performance variables, with the magnitude of the associations shifting across the velocity-time curve as sprint distance increased.

#### 3.4.1. Theoretical Maximal Horizontal Force (F_H0_)

There were large-very large significant negative associations (r = −0.53 to −0.86; *p* < 0.05 to < 0.001) between F_H0_ and sprint split times outcomes ≤ 0–10 m and τ. Large-very large significant positive associations (r = 0.6 to 0.88; *p* < 0.01 to < 0.001) were found between F_H0_ and all momentum outcomes.

#### 3.4.2. Relative Theoretical Maximal Horizontal Force (F_H0rel_)

Large-near perfect significant negative associations (r = −0.52 to −0.98; *p* < 0.05 to <0.001) were found between F_H0rel_ and sprint split-time from 0–2 to 0–40 m, the ∆10–20 m split time and τ.

#### 3.4.3. Theoretical Maximal Horizontal Velocity (v_H0_)

Moderate-near perfect negative significant associations (r = −0.47 to −0.97; *p* < 0.05 to <0.001) were found between v_H0_ and the ∆10–20 m split time and split times > 0–20 m. v_H0_ also had a significant perfect positive association with v_max_ (r = 1.00; *p* < 0.001).

#### 3.4.4. Theoretical Maximal Horizontal Power (P_Hmax_)

Moderate-very large significant negative associations between P_Hmax_ and sprint split times from 0–40 m and τ (r = −0.47 to −0.78; *p* < 0.05 to <0.001). Large-very large significant positive associations between P_Hmax_ and sprint split times from 0–40 m and τ (r = −0.47 to −0.78; *p* < 0.05 to <0.001).

#### 3.4.5. Relative Theoretical Maximal Horizontal Power (P_Hmaxrel_)

Large-near perfect significant negative associations (r = −0.60 to −0.99; *p* < 0.01 to <0.001) were found between P_Hmaxrel_ and all split times across the 40 m sprint and τ. Large significant positive associations (r = 0.51; *p* < 0.05) were also found between P_Hmaxrel_ and v_max_.

#### 3.4.6. Slope of Horizontal Force-Velocity Relationship (S_HFv_)

Moderate-perfect significant positive associations (r = 0.49 to 1; *p* < 0.05 to 0.001) were also found between S_HFv_ and sprint split-times from 0–2 to 0–10 m and τ. A moderate significant negative association (r = 0.49) was found between S_HFv_ and 0–10m momentum.

#### 3.4.7. Rate of Decrease in Ratio of Force with Increasing Velocity during Sprint Acceleration (D_RF_)

Large-near perfect significant positive associations (r = 0.62 to 0.97; *p* < 0.05 to 0.001) were found between D_RF_ and sprint split-times from 0–2 to 0–20 m and τ. A moderate significant negative association (r = 0.51) was found between D_RF_ and 0–10 m momentum.

#### 3.4.8. Maximum Ratio of Step-Averaged Horizontal Ground Reaction Force to the Corresponding Resultant Force (RF_max_)

Large-near perfect significant negative associations (r = −0.53 to −0.98; *p* < 0.05 to <0.001) were found between RF_max_ and sprint split-time from 0–2 to 0–40 m, the ∆10–20 m split time and τ.

#### 3.4.9. Maximum Sprinting Velocity (v_max_)

Large-near perfect significant negative associations (r = −0.63 to −0.98; *p* < 0.01 to <0.001) were found between v_max_ and sprint split-time from 0–20 to 0–40 m, the individual ∆10 m and ∆20 m splits recorded after 10 m.

#### 3.4.10. Maximum Sprinting Acceleration (a_max_)

Trivial-moderate associations (r = −0.34 to 0.06;) were found between mechanical and sprint performance variables, however, all associations failed to reach statistical significance (*p* > 0.05).

#### 3.4.11. Acceleration Relative to a Time Constant (τ)

Moderate-perfect significant positive associations (r = 0.48 to 0.81; *p* < 0.05 to < 0.001) were also found between τ and sprint split-times from 0–2 to 0–20 m. All other horizontal force production variables failed to reach statistical significance (*p* > 0.05) with the sprint performance outcomes.

## 4. Discussion

To our knowledge, this is the first study to compare the jump and sprint mechanical profiles of male academy rugby league players between positions, as well as the investigation of the relationships between jump and sprint mechanical variables and with sprint performance. The findings showed that forwards had greater vertical and horizontal force (large and moderate ES respectively) and sprint momentum (moderate ES) compared to the backs, although non-significant, moderate ES were found for P_Vmaxrel_, P_Hmax_, v_H0_, and v_max_. The backs were significantly faster (moderate ES) than forwards over the ∆10–20 m and ∆30–40 m split times and jumped higher (unloaded, large ES). The academy players’ S_VFv_ relationship differed from the optimal profile in the jump profiles, representing a range from low-high force and low-velocity deficits and well-balanced profiles. When analysing the associations between matched jump and sprint mechanical variables, only trivial-small associations were found. For the associations with sprint performance data, large significant relationships were found between jump *F_V_*_0_ and sprint momentum. Moderate to near-perfect significant relationships (positive and negative) were found between sprint mechanical profiles and sprint performance variables, with the magnitude of the associations shifting across the velocity-time curve as sprint distance increased.

### 4.1. Mechanical Variables in Academy Rugby League Players

#### 4.1.1. Jump Profiles

The mechanical profiles provide the only reference data for academy rugby league players building upon research in senior players [14,15,16]. Jump profiles in senior NRL professional rugby league players [14] (measured using GymAware, Canberra, ACT, Australia) showed greater F_V0rel_ (64.7 ± 16.9 vs. 31.3 ± 5.2 N·kg^−1^) and P_Vmaxrel_ (44.7 ± 6.7 vs. 31.5 ± 9.6 W·kg^−1^) than the current findings. This would be expected as strength and power have been shown to be advantageous to senior performance standards [38,39]. However, previous research [40] has shown using GymAware overestimates jump height compared to force plates, and as such, the differences between profiles are likely smaller. Practitioners and researchers should be aware that the magnitude of the mechanical variables may differ when obtained from other measurement methods (e.g., GymAware linear position transducer [41]). Future research should compare differences between jump mechanical profiles between performance standards using accurate measures of jump height (i.e., optojump or force platforms).

#### 4.1.2. Sprint Profiles

The sprint profiles of academy players presented were lower than super league professional players [16], including F_H0_ (761.8 ± 112.5 vs. 672.3 ± 121.2 N), F_H0rel_ (8.8 ± 1.1 vs. 7.4 ± 1.1 N·kg^−1^), v_H0_ (9.1 ± 0.6 vs. 8.8 ± 0.7 m·s^−1^), P_Hmax_ (1727 ± 277 vs. 1471 ± 246 W) and P_Hmaxrel_ (19.8 ± 2.2 vs. 16.3 ± 9.6 W·kg^−1^). This is consistent with the mean sprint profiles in male international rugby league players [15]. Despite having lower mechanical variables, the mean mechanical efficiency (RF_max_, D_RF_) was greater in the academy players. This may reflect differences in technical capabilities and training exposure in the respective cohorts. As previously reported, youth elite rugby league players appear to have inferior physical capacities (e.g., lower body strength, power, and sprint performance) than senior players respectively [12], however, comparing jump and sprint mechanical profiles may further explain such differences and be a direction for future research.

### 4.2. Positional Comparisons

No previous research has compared the differences between playing positions in mechanical profiles in rugby league. Findings showed forwards produced greater absolute force compared to the backs. Consistent with previous research using strength assessments whereby absolute strength showed differences, but relative measures did not [42,43]. The absolute strength and sprint momentum differences may be apparent due to the positional demands, with forwards being involved in more contact- and collision-based activities during match play, requiring forwards to have greater absolute strength, body mass, and fat-free mass [12,13,44]. As backs display a greater frequency of jumping and sprinting actions than forwards, it could be expected that greater relative force capabilities would be apparent due to the relationships between lower body relative strength, power, and sprint performance [13,23,45]. Backs have consistently been identified as faster and able to jump higher than forwards [12,13]. The backs were significantly faster in between split times only (∆10–20 m and ∆30–40 m). Although non-significant (*p* = 0.06 and *p* = 0.07 respectively), moderate differences were found between playing positions in v_max_ and v_V0_. No other variables reached statistical significance for the differences between positions. Instead, the results suggest the differences in sprint performance in the backs result from greater v_max_ and v_V0_ capabilities, inferring that the backs can apply greater horizontal force at faster velocities. Future studies should assess the positional differences with larger samples and greater statistical power to identify if they find consistent findings.

### 4.3. F-v Imbalances in Profiles

Despite the differences presented between positions, as with previous rugby cohorts’ [3,23], there is clear inter-individual variability for jump and sprint mechanical profiles. A force deficit reflects that the athlete’s optimal load for maximising power during vertical jumping is lower than their own body mass, compared to a velocity deficit where loads higher than their body mass are required respectively [3]. The magnitude of the deficit is based on the difference between optimal load and body mass [3]. There is currently no optimal sprint force-velocity relationship for enhanced sprint performance; instead, it appears separate capacities underpin performance [2,23,45]. A force deficit was most common in this cohort’s jump mechanical profiles, representing a range of high (n = 10) and low (n = 5) force deficits. Only forwards presented low (n = 2) velocity deficits and well-balanced (n = 1) profiles. This contrasts with Samozino et al.,’s [3] findings in senior international rugby players. The differences in profiles may be a result of several factors including the chronic exposure to training and game demands, age, maturity, and resistance training experience to produce different mechanical profiles over time [20,46,47]. The large range of profiles evident within these rugby league athletes, combined with previous sports findings, indicates that such variables are more individual than sport-specific [23,45]. Therefore, by taking the athletes’ mechanical profile into consideration, a group approach could be replaced by individualised training (i.e., reducing imbalances and increasing performance [1,48]).

### 4.4. Associations between Jump Mechanical and Sprint Variables

No significant associations between the matched jump and sprint mechanical variables suggest that they provide distinctive information about the athlete’s capacities, supporting the task-dependency of athlete’s jump and sprint mechanical profiles in “elite” or higher standard populations [22,23,24]. This is further highlighted in the non-significant, trivial-moderate associations between jump mechanical variables and sprint performance. The absence of association between jump and sprint variables supports previous research [22,23], suggesting caution is needed when inferring changes in vertically orientated training (e.g., strength and power training) to improvements in sprinting mechanical and performance outcomes, especially in academy athletes [22,23]. Considering that after as little as two years of systematic resistance training (e.g., u16’s strength and power training) there are only trivial-moderate non-significant associations with enhanced sprint performance outcomes, large improvements would likely be required to meaningfully enhance the athlete’s performance [49]. This is particularly important as this is the first study to explore the relationships between jump and mechanical variables and sprint mechanical and performance variables in an elite youth population.

The mechanical constraints of sprinting (i.e., contact times shorter than time to achieve peak force [50]) and differences in segmental sequencing may reduce the transfer to horizontal variables, particularly in athletes with presumably superior physical capabilities in “elite” populations [12,13]. In the sprinting action, the mechanical variables reflect the ability to effectively apply horizontal force into the ground at progressively increasing velocities (mechanical effectiveness). Sprinting involves successive eccentric and concentric muscle actions closer to a counter-movement jump rather than just a solely concentric action (i.e., squat jump) [51]. Therefore, the lack of association in the jump mechanical profile may suggest greater importance in using the stretch-shortening cycle (pre-stretch augmentation) in dynamic fast contractions (i.e., <0.25 s) such as the sprint action [52]. Hence, improvements may instead be represented by the ability to effectively apply horizontal force into the ground at progressively increasing velocities (mechanical effectiveness) and to utilise the stretch-shortening cycle (pre-stretch augmentation) [2,53]. Hence, sprint development in this population may require greater mechanical specificity and specific interventions to enhance athlete’s segmental sequencing, mechanical and fast stretch-shortening capabilities. Further research is required to explore the influence of training age and baseline physical qualities coinciding with the reduction of training “transfer”.

The large positive significant associations which were found between the jump F_V0_ and sprint momentum highlight the importance of absolute force production often associated with greater lean mass. Research has shown body mass increases as players move into senior rugby league, yet the average sprint times are faster [54,55]. Therefore, practitioners should look to increase body mass and mechanical capacities that positively influence sprinting ability (i.e., force, velocity, power) concurrently to prevent reductions in sprint performance [56].

### 4.5. Associations between Sprint Mechanical Variables and Sprint Performance

Consistent with previous research, our findings reported moderate to near-perfect significant associations between sprint mechanical variables and sprint performance, with the strength of the relationships shifting across the velocity-time curve from short-sprints (F_H0rel_ and RF_max_) to longer distance sprints (v_max_, v_H0_, and D_RF_) [57,58]. Athletes possessing a greater F_H0rel_, RF_max_, and a more force orientated S_HFv_, were associated with the sprint time in sprint segments < 20 m. This relationship implies that the greater an athlete’s ability to apply high forces relative to body mass and, oriented in an anteroposterior direction, the greater their short-sprint performance (i.e., 0–20 m). Athletes that possessed greater *v_H_*_0_, *D_RF_*, and velocity orientated S_HFv_ were better associated with the sprint time in sprint segments > 20 m. The S_HFv_ and P_max_ appear to be a performance indicator of both short and long distance sprints because their computations are based on F_H0_ and v_H0_. Consistent with previous research, P_Hmaxrel_ displayed the strongest association with faster time at each of the sprint split’s > 0–5 m [57,58]. However, when evaluated using between sprint split times, the magnitude of the association decreases following 10 m. This is likely a result of P_Hmaxrel_ occurring in the first few sprint acceleration steps and the gradual shift in the orientation of the anterior-posterior ground reaction force vector becoming more vertically orientated as the athlete transitions to more upright v_max_ running positions [59,60]. Therefore, these results show that variations in an athlete’s sprint mechanical profile can explain some of the individual differences in performance for short to medium-long distance sprints. Therefore, practitioners may individualise sprint development contents based on players’ mechanical variables while supported by individual match play requirements.

The results showed that absolute maximum capabilities F_H0_ and P_Hmax_ were all largely-very largely positively associated with superior momentum outcome measures. An athlete with a larger body mass requires greater absolute force and power to perform the same sprint velocity/time. This is important as athletes with a higher mass, and similar sprint capacities will possess higher momentum (momentum = mass × velocity) and are more successful in contact- and collision-based activities during match play within academy rugby league [61]. Hence, athletes who can produce higher absolute force and power are in an advantageous position compared to lighter individuals. Therefore, athletes looking to increase momentum might emphasise training to increase skeletal muscle mass to increase body mass in a position-specific manner (i.e., greater focus in forwards) due to the associations with absolute force production capabilities. However, more mass is not necessarily advantageous without a concurrent increase in relative force and power. Moderate-large negative associations were identified for RF_max_, D_RF_, τ, and v_H0_ for 0–10 m momentum. These findings suggest that athletes with greater momentum (typically heavier athletes) were associated with lower mechanical efficacy, slower horizontal velocity capabilities, and they reached a greater percentage of v_max_ more quickly (typically associated with athletes with a slower v_max_ [62]). As an athlete gets heavier, the energy cost of accelerating that mass also increases, as does the aerodynamic drag associated with pushing that wider frontal area through the air [63]. Practitioners should consider whether the advantage of higher momentum from an increase in body mass is worth a possible detrimental effect on sprint performance outcomes.

Our findings identified that the time taken to reach a high percentage of v_max_ (τ) presented very large (splits ≤ 0–10 m) to moderate (0–20 m) significant associations with slower sprint performance in short distance sprint outcomes only. The a_max_ presented no significant association with any sprint performance outcome. In contrast, the magnitude of the v_max_ achieved presented large to near-perfect associations (at splits > 10 m), with faster sprint times in both short and medium-long distance sprints. There appears to be a cross-over of the importance of the time taken to reach a high percentage of v_max_ in a shorter time and the magnitude of the v_max_ achieved at between 10 and 20 m with the magnitude of the v_max_ having a greater association from the ∆10–20 m split and 0–20 m outcomes. The associations between v_max_ and sprint split time are consistent with previous research identifying that an athlete’s v_max_ had very large to near-perfect associations with sprint outcomes (9.1, 18.3, and 36.6 m times). Research has shown that irrespective of sprinting times, both “fast” and “slow” athletes accelerated in a similar pattern relative to v_max_ [26,62]. These results indicate that the v_max_ serves as a limiting factor to performance, and a higher v_max_ may enable a superior acceleration phase and short-distance sprint performance [62]. Thus, highlighting that v_max_ is of critical importance to the sprint outcomes in rugby league athletes particularly as most sprints initiate from a moving start [64]. Therefore, these findings do not discount the common suggestion that acceleration phase outcomes should be the primary focus, due to the sprint distance in rugby league frequently being <20 m. Instead, we suggest that greater inclusion of v_max_ training may be warranted for all athletes to enhance each of the different phases of a linear sprint with greater relative importance attributed based on positional requirements [62,64].

### 4.6. Limitations

These findings constitute novel measurement, analysis, and evaluation tools for academy rugby league players built upon existing research presenting these athletes’ physical qualities (e.g., [12]). As the data only represents a single team, the relatively small sample size will have affected the study’s statistical power, meaning an increased likelihood of smaller differences not reaching statistical significance (type 2 error). Further experimental research is therefore required with greater sample sizes to confirm our findings and to identify whether the findings can be extrapolated to training-induced effects and the potential transfer between, for instance, jumping-type training and sprinting performance and to elite youth populations in same and alternative sports (i.e., soccer). Association, in this case, does not necessarily imply causation; for example, the strong relationship between v_max_ and short sprint performance does not necessarily indicate that v_max_ determines acceleration phase performance. It is important to note that for athletes who do not perform the sprint start and acceleration phase with a requisite amount of motor skill, overall sprint performance will be negatively affected, regardless of the athlete’s v_max_. In the present study, we used the field methods proposed by Samozino and colleagues to determine the mechanical profiles in jumping and sprinting [27,28,29]. It is important to note that both the sprint mechanical profiles and performance variables were derived for the processed instantaneous velocity-time-position data from the fastest 40 m trial which may to an extent explain the strength of the associations presented. Despite both the jump and sprint mechanical profiles previously showing high validity compared to the force plate method (gold-standard), it is plausible that a force platform could have provided more accurate data due to methodological limitations associated with the field methods of mechanical profiling [27,28,29]. However, the results of this study present practical interest because Samozino’s methods can be implemented in practice by many physical coaches, while the use of force plates could be limited to laboratory conditions.

## 5. Conclusions

The current study presented the mechanical profiles of academy rugby league players for the first time. Forwards had significantly greater mechanical variables (F_V0_ and F_H0_,) and sprint momentum, whilst backs were significantly faster and jumped higher (unloaded). The jump S_VFv_ differed from the optimal profile across both positions, representing low-high force, low-velocity deficits, and well-balanced profiles. Although positional differences in mechanical variables were apparent, the mechanical capabilities are more individual than position specific. No significant associations between the matched jump and sprint mechanical variables indicate that they provide distinctive information regarding the lower limb’s mechanical capabilities. Therefore, both jump and sprint profiling appear to present utility for individualised training. Furthermore, associations between sprint mechanical variables and sprint performance are dependent upon sprint distance. However, developing horizontal power relative to body mass seems key to enhance sprint performance. The secondary focus should be placed on targeting individual force and velocity and mechanical capabilities. These findings will help practitioners inform their training decisions with targeted training prescription, player profiling, and monitoring.

## Figures and Tables

**Figure 1 sports-09-00093-f001:**
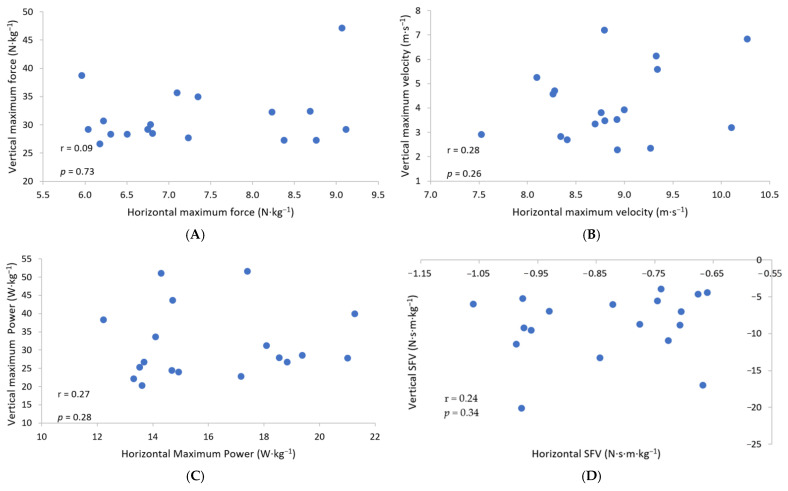
Association between matched force-velocity relationship variables obtained from the jump and sprint mechanical profiles (**A**) theoretical maximal force (F_0_); (**B**): theoretical maximal velocity (v_0_); (**C**): theoretical maximal mechanical power (P_max_); (**D**): the slope of the linear F-v relationship. r = correlation coefficient, *p* = *p*-value.

**Table 1 sports-09-00093-t001:** Jump and sprint mechanical profiles of academy rugby league players.

Variables	All (n = 20)	Playing Position		Between Position Comparison	
		Backs (n = 7)	Forwards (n = 13)	ES 90% CI	*p*-Value
Unloaded jump height (cm)	34.8 ± 3.4	36.6 ± 3.8	33.7 ± 2.6	0.96 (0.05 to 1.77)	0.04
Jump Vertical Mechanical Variables					
F_V0_ (N)	2869 ± 579	2426 ±218	3091 ± 580	−1.34 (−2.17 to −0.38)	0.01
F_V0rel_ (N·kg^−1^)	31.3 ± 5.2	31 ± 2.4	31.5 ± 6.2	−0.11 (−0.93 to 0.72)	0.43
v_V0_ (m·s^−1^)	4.16 ± 1.51	4.69 ± 1.71	3.89 ± 1.41	0.53 (−0.33 to 1.34)	0.35
P_Vmax_ (W)	2874 ± 889	2812 ± 981	2904 ± 884	−0.10 (−0.92 to 0.73)	0.64
P_Vmaxrel_ (W·kg^−1^)	31.5 ± 9.6	35.5 ± 11.3	29.5 ± 8.4	0.63 (−0.24 to 1.44)	0.24
S_VFv_ (N·s·m·kg^−1^)	−8.80 ± 4.42	−7.56 ± 3.22	−9.42 ± 4.92	0.42 (−0.43 to 1.23)	0.45
Jump F-v Imbalances					
Jump Fv_imb_ (%)	58.4 ± 29.6	50.3 ± 25.1	62.4 ± 31.9	−0.40 (−1.21 to 0.44)	0.40
High force deficit	n = 10	n = 3	n = 7		
Low force deficit	n = 5	n = 3	n = 2		
Well balanced	n = 1	n = 0	n = 1		
Low velocity deficit	n = 2	n = 0	n = 2		
High velocity deficit	n = 0	n = 0	n = 0		
Sprint Horizontal Mechanical Variables					
F_H0_ (N)	672.3 ± 121.2	596.4 ± 113.2	713.1 ± 108	−1.06 (−1.83 to −0.20)	0.045
F_H0rel_ (N·kg^−1^)	7.39 ± 1.08	7.42 ± 1.07	7.37 ± 1.13	0.05 (−0.72 to 0.82)	0.92
v_H0_ (m·s^−1^)	8.80 ± 0.65	9.23 ± 0.76	8.57 ± 0.46	1.16 (0.28 to 1.93)	0.06
P_Hmax_ (W)	1471 ± 245	1370 ± 238	1526 ± 241	−0.65 (−1.41 to 0.17)	0.18
P_Hmaxrel_ (W·kg^−1^)	16.3 ± 2.7	17.1 ± 2.5	15.8 ± 2.8	0.48 (−0.47 to 1.39)	0.31
S_HFv_ (N·s·m·kg^−1^)	−0.84 ± 0.13	−0.81 ± 0.15	−0.86 ± 0.13	0.37 (−0.32 to 1.24)	0.49
Mechanical efficiency					
RF_max_ (%)	0.50 ± 0.04	0.51 ± 0.04	0.50 ± 0.04	0.21 (−0.58 to 0.97)	0.43
D_RF_ (%)	−0.08 ± 0.01	−0.07 ± 0.01	−0.08 ± 0.01	0.46 (−0.34 to 1.22)	0.48
Velocity trace characteristics					
v_max_ (m·s^−1^)	8.51 ± 0.58	8.88 ± 0.67	8.31 ± 0.43	1.11 (0.25 to 1.88)	0.07
a_max_ (m·s^−2^)	7.46 ± 0.99	7.78 ± 1.26	7.28 ± 0.82	0.51 (−0.30 to 1.27)	0.38
τ (s)	1.17 ± 0.18	1.22 ± 0.20	1.15 ± 0.17	0.37 (−0.42 to 1.14)	0.45
Split Times					
0–2 m time (s)	0.82 ± 0.06	0.80 ± 0.05	0.82 ± 0.06	−0.33 (−1.09 to 0.46)	0.52
0–5 m time (s)	1.41 ± 0.08	1.38 ± 0.08	1.41 ± 0.09	−0.35 (−1.11 to 0.44)	0.50
0–10 m time (s)	2.15 ± 0.12	2.12 ± 0.11	2.17 ± 0.13	−0.40(−1.16 to 0.39)	0.40
0–20 m time (s)	3.46 ± 0.18	3.38 ± 0.16	3.50 ± 0.19	−0.66 (−1.43 to 0.15)	0.18
0–30 m time (s)	4.68 ± 0.24	4.57 ± 0.21	4.73 ± 0.24	−0.69 (−1.46 to 0.13)	0.14
0–40 m time (s)	5.87 ± 0.30	5.71 ± 0.27	5.96 ± 0.29	−0.88 (−1.65 to −0.04)	0.09
Between split times					
∆10–20 m time (s)	1.31 ± 0.07	1.26 ± 0.06	1.33 ± 0.07	−1.05 (−1.82 to −0.19)	0.04
∆20–30 m time (s)	1.22 ± 0.07	1.18 ± 0.07	1.24 ± 0.06	−0.94 (−1.71 to −0.10)	0.10
∆30–40 m time (s)	1.20 ± 0.08	1.15 ± 0.08	1.22 ± 0.07	−0.95 (−1.72 to −0.11)	0.03
Momentum					
0–10 m Momentum (kg·s^−1^)	424 ± 52	379 ± 40	448 ± 40	−1.73 (−2.53 to −0.77)	0.001
30–40 m Momentum (kg·s^−1^)	761 ± 275	699 ± 61	794 ± 61	−1.56 (−2.35 to −0.63)	0.004

Data presented as mean ± the standard deviation or frequency (n) and range for the F-v imbalance descriptors: High force deficit ≤ 60%, low force deficit = 60–90%, well balance ≥ 90–110%, low velocity deficit = 110–140% and high velocity deficit ≥ 140%. Abbreviations: a_max_ = maximum acceleration; BM = body mass; F_0_ = theoretical maximal force; F_0rel_ = relative theoretical maximal force; F-v deficit = magnitude of the difference between actual and optimal F-v profiles; H = horizontal; P_max_ = theoretical maximal power; P_maxrel_ = relative theoretical maximal power; S_Fv_ = slope of force-velocity relationship; SJ, squat jump; V = vertical; v_0_ = theoretical maximal velocity; v_max_ = maximum sprinting velocity; 90% CI = 90% confidence intervals.

**Table 2 sports-09-00093-t002:** Associations between jump mechanical variables and sprint performance outcomes.

Performance Outcomes	F_V0_ (N)	F_V0rel_ (N·kg^−1^)	v_V0_ (m·s^−1^)	P_Vmax_ (W)	P_Vmaxrel_ (W·kg^−1^)	S_VFv_ (N·s·m·kg^−1^)
Split Times						
0–2 m (s)	−0.10	−0.18	0.15	−0.01	−0.07	0.17
0–5 m (s)	−0.07	−0.17	0.11	−0.04	−0.10	0.14
0–10 m (s)	0.02	−0.18	0.07	−0.08	−0.20	0.07
0–20 m (s)	0.17	−0.26	−0.01	−0.10	−0.32	0.01
0–30 m (s)	0.14	−0.30	−0.05	−0.12	−0.32	0.01
0–40 m (s)	0.20	−0.30	−0.10	−0.15	−0.38	−0.05
Between split time						
∆10–20 m (s)	0.27	−0.29	−0.13	−0.14	−0.41	−0.07
∆20–30 m (s)	0.24	−0.34	−0.17	−0.10	−0.34	−0.02
∆30–40 m (s)	0.39	−0.35	−0.24	−0.06	−0.36	−0.07
Momentum						
Mom@ 0–10m (kg·s^−1^)	0.71 **	−0.16	−0.25	0.18	−0.27	−0.19
Mom@ 30–40m (kg·s^−1^)	0.75 **	0.06	−0.11	0.17	−0.28	−0.25
Velocity trace characteristics						
v_max_ (m·s^−1^)	−0.18	0.31	0.26	0.11	0.40	0.10
a_max_ (m·s^−2^)	−0.31	−0.01	0..29	0.12	0.36	0.23
τ (s)	−0.29	0.03	0.30	0.00	0.13	0.20

Note: a_max_ = maximum acceleration; F_V0_ = theoretical maximal vertical force; F_V0rel_ = relative theoretical maximal vertical force; Fv_imb_ = magnitude of the difference between actual and optimal F-v profiles; P_Vmax_ = theoretical maximal vertical power; P_Vmaxrel_ = relative theoretical maximal vertical power; S_VFv_ = slope of vertical force-velocity relationship; SJ = squat jump; v_V0_ = theoretical maximal vertical velocity. Significant associations: * *p* < 0.05, ** *p* < 0.01, *** *p* < 0.001.

**Table 3 sports-09-00093-t003:** Associations between sprint mechanical variables and sprint performance outcomes.

Performance Outcome Variables	F_H0_ (N)	F_H0__rel_ (N·kg^−1^)	v_H0_ (m·s^−1^)	P_Hmax_ (W)	P_Hmaxrel_ (W·kg^−1^)	S_HFv_ (N·s·m·kg^−1^)	D_RF_ (%)	RF_max_ (%)	v_max_ (m·s^−1^)	a_max_ (m·s^−2^)	τ (s)
Split Times											
0–2 m (s)	−0.63 **	−0.98 ***	−0.18	−0.78 ***	−0.94 ***	0.77 ***	0.78 ***	−0.98 ***	−0.23	−0.02	0.81 **
0–5 m (s)	−0.60 **	−0.96 ***	−0.28	−0.78 ***	−0.98 ***	0.74 ***	0.72 ***	−0.97 ***	−0.34	0.06	0.77 **
0–10 m (s)	−0.53 *	−0.94 ***	−0.38	−0.74 ***	−0.99 ***	0.65 **	0.62 **	−0.94 ***	−0.43	−0.01	0.70 **
0–20 m (s)	−0.36	−0.83 ***	−0.58 **	−0.66 **	−0.99 ***	0.43	0.40	−0.84 ***	−0.63 **	−0.07	0.48 *
0–30 m (s)	−0.22	−0.71 **	−0.72 **	−0.57 **	−0.94 ***	0.34	0.31	−0.72 ***	−0.76 ***	−0.11	0.39
0–40 m (s)	−0.10	−0.60 **	−0.81 ***	−0.47 *	−0.88 ***	0.24	0.19	−0.61 **	−0.85 ***	−0.13	0.28
Between split time											
∆10–20 m (s)	−0.02	−0.52 *	−0.86 ***	−0.41	−0.84 ***	0.14	0.11	−0.53 *	−0.89 ***	−0.17	0.19
∆20–30 m (s)	0.20	−0.25	−0.96 ***	−0.21	−0.63 **	−0.24	−0.27	−0.26	−0.97 ***	−0.19	−0.20
∆30–40 m (s)	0.30	−0.19	−0.97 ***	−0.11	−0.60 **	−0.35	−0.37	−0.21	−0.98 ***	−0.17	−0.30
Momentum											
Mom 0–10m (kg·s^−1^)	0.88 ***	0.33	−0.47 *	0.75 ***	0.09	−0.49 *	−0.51 *	0.32	−0.43	−0.34	−0.51 *
Mom@ 30–40m (kg·s^−1^)	0.60 **	0.00	−0.15	0.57 **	−0.06	0.07	0.03	0.00	−0.14	−0.32	−0.04
Velocity trace characteristics											
v_max_ (m·s–^1^)	−0.35	0.09	1.00 ***	0.06	0.51 *	0.36	0.37	0.11			
a_max_ (m·s–^2^)	−0.24	−0.9	0.18	−0.22	0.05	0.14	0.08	−0.04			
τ (s)	−0.86 ***	−0.87 ***	0.36	−0.79 ***	−0.60 **	1.00 ***	0.97 ***	−0.80 ***			

Note: a_max_ = maximum acceleration; D_RF_ = rate of decrease in ratio of force with increasing velocity during sprint acceleration; F_H0_ = theoretical maximal horizontal force; F_H0rel_ = relative theoretical maximal horizontal force; P_Hmax_ = theoretical maximal horizontal power; P_Hmaxrel_ = relative theoretical maximal horizontal power; RF_max_ = maximum ratio of step-averaged horizontal ground reaction force to the corresponding resultant force; S_HFv_ = Slope of horizontal force-velocity relationship; τ = acceleration relative to a time constant; v_H0_ = theoretical maximal horizontal velocity; v_max_ = maximum sprinting velocity. Significant associations: * = *p* < 0.05, ** = *p* < 0.01, *** = *p* < 0.001.

## Data Availability

The datasets generated during and/or analysed during the current study are available from the corresponding author on reasonable request.

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
