# Peer review of "Sprint and Jump Mechanical Profiles in Academy Rugby League Players: Positional Differences and the Associations between Profiles and Sprint Performance"

_sports, 2021, doi:10.3390/sports9070093_

Round 1

Reviewer 1 Report

The authors developed a cross-sectional research design where the jump and sprint mechanical profiles were assessed in academy rugby league players. This study is a novelty because most studies conducted until now did not address these young rugby samples.

The paper is very well constructed and address critical issues for rugby training preparation aiming at the senior level.

However, in the reviewer opinion, some minor aspects could improve the manuscript.

Introduction

This section is well-present and clarifies the aims and pertinence of the study.

Methods

The usual training schedule, namely the strength and sprint training sessions or tasks, should have a little more detail in the reviewer opinion.

The warmup schedule needs to be better described because the recovery time between repetitions and series is a solid criterion to avoid unnecessary fatigue. Please add the request information

During Squat jumps, how did the authors control the knee angle (∼90-100â—¦ knee angle)?

What equipment was used in the jump performance assessment? Please add.

The equations used for calculations could be present in the manuscript and not only referenced.

As a reliability index, the authors reported de CV.

Why not report other intra and inter reliability indexes like ICC and the Typical error? In the reviewer opinion, it will be welcome. The same advice is applied to sprint testing procedures.

The overall statistical approach is adequate.

Results

Comparing positions authors highlighted the main differences founded between forwards and backs on the jump and speed performance variables

Discussion

Very well presented. The authors used their results to interpret the relationship between jump and sprint capacities in league rugby players, highlighting these motor abilities' specificity. In the reviewer opinion, the discussion with the published literature is appropriately done.

The limitation section is an essential reflection for future research on the topic.

The reviewer agrees with the authors to include the conclusion section, overwhelming the long and complex discussion section.

Reviewer 2 Report

STRUCTURE

  • The manuscript is properly structured, although the typography and the sequence of the paragraphs should be revised (e.g. paragraph 4.5. is missing)

TITLE AND ABSTRACT

  • The title or abstract should inform that the type of study

INTRODUCTION

  • Elaborate further the importance of these parameters in rugby. A recently article published in this journal that may shed light on this comment is:

Zabaloy S, Carlos-Vivas J, Freitas TT, Pareja-Blanco F, Pereira L, Loturco I, Comyns T, Gálvez-González J, Alcaraz PE. Relationships between Resisted Sprint Performance and Different Strength and Power Measures in Rugby Players. Sports (Basel). 2020 Mar 14;8(3):34. doi: 10.3390/sports8030034. PMID: 32183262; PMCID: PMC7183066.

  • In the introduction it says "To date, no study in rugby 57 league has included both methods in one cohort, profiled academy rugby 58 league players or provided comparisons between playing positions", however, as it is also narrated in the article itself (lines 86-89) there are data in senior teams, although without differentiation of position. A research on sprinting and jumping strength in senior players that would be interesting to consider is:

Loturco I, Pereira LA, Moraes JE, Kitamura K, Cal Abad CC, Kobal R, Nakamura FY. Jump-Squat and Half-Squat Exercises: Selective Influences on Speed-Power Performance of Elite Rugby Sevens Players. PLoS One. 2017 Jan 23;12(1):e0170627. doi: 10.1371/journal.pone.0170627. PMID: 28114431; PMCID: PMC5256944.

  • State the main and specific objectives clearly
  • Add the research hypotheses

MATERIAL AND METHODS

Study design

  • Present key elements of study design early in the paper

Setting

  • Describe the setting, locations, and relevant dates, including periods of recruitment, exposure, follow-up, and data collection

Participants

  • The sample number selected is too small to obtain extrapolable results
  • Line 135: This information does not correspond to the section on participants, but rather to the section on procedures
  • Participants were instructed to prepare 135 themselves as they would for regular competition (i.e., nutrition, hydration, sleep and physical activity, with no high-intensity training) 24 hrs 137 before testing.” Was there any way to control these variables?
  • Add the number and date of the ethics committee

Procedures

  • Line 147 to 156: no explanation of who is leading the warm-up, nor the time of the warm-up (i.e. how many sprints they do, how many polymetric jumps, etc.)
  • Line 228: Add reference

Bias

  • Describe any efforts to address potential sources of bias
  • Clearly define all potential confounders, and effect modifiers

Study size

  • How sample size was determined?

Statistical methods

  • Explain how missing data were addressed

RESULTS

Some basic information is missing:

  • Report numbers of individuals at each stage of study—eg numbers potentially eligible, examined for eligibility, confirmed eligible, included in the study, completing follow-up, and analysed
  • Give characteristics of study participants (eg demographic, clinical, social) and information on exposures and potential confounders
  • Table 1: Add "n" next to each group (all, forwards and back)
  • The typography of the numbers in table 1 must be the same as throughout the document
  • Table 3: explain what is meant by the bold letters

DISCUSSION

  • Discuss limitations of the study, taking into account sources of potential bias or imprecision, especially the size of the sample
  • Discuss the generalisability (external validity) of the study results

CONCLUSIONS

  • This section is not mandatory but can be added to the manuscript if 714 the discussion is unusually long or complex” Delete it
  • Conclusions are too long and not very specific

REFERENCES

  • References follow the indicated style

Reviewer 3 Report

Thank you for the opportunity to review this manuscript, which considers some interesting, applied issues. This study appears to be novel, but as submitted needs considerable work on the presentation. The authors showed an interesting point about the “Sprint and Jump Mechanical Profiles in Academy Rugby League Players: Positional Differences and the Associations Between Profiles and Sprint Performance”, unfortunately, there are several points to overcome.

Abstract

  • Please add the “p” value. When the relationships reported

Method section

Line 126: please change the time in (4.00-5.30 p.m.)

Line 173 and 193: please to include the ICC about the reliability  

The figure should be updated (to delete decimal for y and x axes) please recheck each figure for a better presentation in 600 DPI

I suggest ANOVA to compare each group (profile) / in statistical section is missing this information

Round 2

Reviewer 2 Report

Please, include ES values at results. 

Review p or P values in the legend of the tables. 

Conclusions are too long yet, please, modify. 

Reviewer 3 Report

the main document was well improved

Author Response

Thank you!